# Mesenchymal Stem Cell Therapy in Alopecia Areata: Visual and Molecular Evidence from a Mouse Model

**DOI:** 10.3390/ijms25179236

**Published:** 2024-08-26

**Authors:** Song-Hee Park, Seo-Won Song, Yu-Jin Lee, Hoon Kang, Jung-Eun Kim

**Affiliations:** Department of Dermatology, Eunpyeong St. Mary’s Hospital, College of Medicine, The Catholic University of Korea, Seoul 03312, Republic of Korea; saccharide@hanmail.net (S.-H.P.); songseowon@hanmail.net (S.-W.S.); cindyeujine1@naver.com (Y.-J.L.); johnkang@catholic.ac.kr (H.K.)

**Keywords:** alopecia areata, mesenchymal stem cells, immunosuppressive, local, systemic, in vivo, mouse model, treatment

## Abstract

Recent studies have highlighted the potential of Mesenchymal Stem Cells (MSCs) as an alternative treatment for Alopecia Areata (AA) due to their immunosuppressive properties. While MSCs have shown promise in cell experiments, their effectiveness in vivo remains uncertain. This study aims to validate local administration of MSC therapy’s efficacy in AA treatment through animal experiments. AA was induced through Interferon-gamma (IFN-γ) administration in mice, and MSC treatment (MSCT)’s effects were assessed visually and through tissue analysis. The MSC-treated group showed more hair regrowth compared to the control (CTL) group. MSCT notably reduced local inflammatory cytokines (JAK1, JAK2, STAT1, STAT3, IFN-γR, IL-1β, IL-16, IL-17α, and IL-18) in AA-induced mice’s skin, but systemic cytokine levels remained unchanged. Furthermore, MSC treatment normalized the expression of Wnt/β-catenin signaling pathway genes (LEF1 and β-catenin) and growth factors (FGF7 and FGF2), which are crucial for hair cycle regulation. This study lays the groundwork for further exploring MSCs as a potential treatment for AA, but more research is needed to fully understand their therapeutic potential.

## 1. Introduction

Alopecia Areata (AA) is a common non-scarring alopecia with patch-type hair loss. It usually involves scalp hairs, but some patients suffer losses of eyebrow, eyelash, or any other type of body hair. Its prevalence can vary by region and population, but the global prevalence is estimated to be around 0.1% to 0.2% of the general population at any given time, with a lifetime risk of about 1.7% [1,2]. While AA is typically not life-threatening, its impact on patients with extensive scalp hair loss or facial hair involvement is significant, often causing extreme emotional stress [3].

AA is understood as an autoimmune disease, which stems from the breakdown of immune privilege in hair follicles [4,5]. ULBP3, a NKG2D ligand, is massively upregulated in the hair follicles of AA patients, which leads to the collapse of immune privilege, and the sustained recruitment of cytotoxic CD8+NKG2D+T cells around the hair follicles in the hair loss patches [6]. AA is often associated with other autoimmune conditions, such as thyroid disease and vitiligo. Patients with AA are more likely to have a family history of autoimmune diseases [7].

Topical or local injection of corticosteroids, immunotherapy with diphenylcyclopropenone (DPCP), and intermittent or continuous systemic administration of immunosuppressants such as cyclosporine and methotrexate have been used for AA treatment for decades [8]. However, these treatments have a low level of evidence, and long-term use of immunosuppressive agents can cause side effects such as high blood pressure and liver or kidney damage.

Numerous endeavors have explored potential AA-targeted treatments. Currently, the only FDA-approved AA treatment is Janus kinase (JAK) inhibitors [9]. Despite these recent advances, treatment-resistant cases still exist, and most responders to JAK inhibitors experience relapses or exacerbating symptoms after discontinuation or dose reduction. Moreover, a significant portion of AA patients are young adults, and the long-term safety of JAK inhibitors is not well known. These unmet needs in AA treatment highlight the need for more effective and less cytotoxic alternatives such as stem cell therapy.

There are various sources of mesenchymal stem cells (MSCs) including the skin fibroblasts, peripheral blood, bone marrow, adipose tissue, and tonsils. Regardless of their origins, MSCs have immunosuppressive effects; thus, administration of allogenic MSCs does not result in immune rejection [10]. For this reason, the use of allogenic or autologous MSCs has been explored in refractory autoimmune diseases such as graft-versus-host disease and rheumatoid arthritis, and they have shown promising results [11,12,13]. 

There is some cumulative in vitro and ex vivo evidence that MSCs could be potential therapeutic alternatives in AA treatment. Byun et al. reported that MSC pretreatment prevented AA induction in mouse models [14]. Li et al. suggested that AA patients showed sustained hair regrowth after one-time exposure to MSCs via extracorporeal circulation of their peripheral blood. So called ‘stem cell education therapy’ lead to hair regrowth which was maintained for up to 1–2 years. Expression of TGF-β in the hair follicles was increased after the exposure to stem cells [15]. 

Our previous experiments demonstrated MSCs’ immunomodulatory effects on dermal papilla and outer root sheath cells, enhancing hair follicle immune privilege (HF-IP) maintenance, especially pre-Interferon-gamma (IFN-γ) treatment [11,16,17]. MSCs prevented catagenic change in an ex vivo hair follicle (HF) organ culture model and enabled hair length growth. MSCs strongly induced immune-tolerance effects when cocultured with peripheral blood mononuclear cells of severe AA patients. However, there is still a lack of in vivo evidence that MSC treatment (MSCT) is beneficial in AA treatment and what its mechanism of action is.

We aimed to evaluate the effectiveness of locally administered MSCs in an animal model of AA. Our hypothesis is that MSCT in an AA-induced mice model will reduce local inflammation and improve AA symptoms. 

## 2. Results

### 2.1. AA Induction

There is an absence of AA mice models via gene modification. We conducted this experiment to explore the therapeutic potential of MSCs in mice with AA induced via IFN-γ. 

Of the 45 mice subjected to AA induction, AA was successfully induced in 35 mice, yielding an induction success rate of approximately 77%. At week 4, of all mice with hair loss, three mice were selected as representatives. They were then sacrificed for histological analysis at the site of hair loss. Increased CD4+CD8+ T cell infiltration in the dermis and subcutaneous perifollicular area was detected, and dystrophic hair follicles were also observed. Based on these findings, we confirmed AA induction. Among the 35 mice in which AA was induced, 21 were allocated to the MSCT group, while 14 were assigned to the control (CTL) group (Appendix A). Nine mice from each group were selected as representatives (Figure 1).

### 2.2. Global Assessment of Efficacy of MSCT in AA Mice 

We captured images of the dorsal skin of AA mice weekly, starting from Day 0, when induction with IFN-γ began. The AA-induced subjects were divided into two groups: the control (CTL) group, which received saline administration, and the MSCT group, which received MSC administration. 

On Day 28, following the observation of extensive AA patches after epilation in both groups, the first MSC injections were administered to the MSC group, with a second round of MSC injections performed on Day 35. By Day 49, hair regrowth was evident in both the MSC-treated and CTL groups. However, hair shedding occurred again in the subsequent hair cycle, with significant hair loss observed in both groups on Day 56, immediately after epilation (Figure 1, Appendix A).

To facilitate a clear comparison of treatment effects, we highlighted areas affected by AA in white and hair-covered areas in black for each mouse at week 4 and 10 (Figure 2). We calculated the proportion of hair-bearing areas via pixel count, defining diffuse AA as areas with less than 25% hair coverage and partial AA as areas with more than 25% hair coverage. At week 4, 8 of 14 mice in the CTL group (C1, C4, C7, and C8–C12) had diffuse AA, and 6 mice (C2, C3, C5, C6, C13, and C14) had partial AA. Similarly, 10 of 21 mice in the MSC group (M1, M4, M6, M9, M12, and M17–M21) had diffuse AA, while 11 mice (M2, M3, M5, M7, M8, M10, M11, and M13–M16) had partial AA. Mice with partial AA at week 4 maintained partial AA at week 10 in both groups. In the CTL group, 2 of 8 mice with diffuse AA (C8 and C9) continued to have diffuse AA at week 10, while the remaining 6 improved to partial AA. In contrast, all mice with diffuse AA in the MSC group at week 4 improved to partial AA by week 10.

We calculated the differences in the proportion of hair-bearing areas between week 4 and 10 for each mouse, and these differences were compared between the CTL and MSC groups (Figure 3). All mice from the CTL and MSC group were included in this comparison. The MSC group showed a significant improvement in hair regrowth compared to the CTL group at week 10. Further analysis revealed that the MSC group with diffuse AA exhibited a significant improvement in hair regrowth compared to the CTL group with diffuse AA at week 10, as did the MSC group with partial AA compared to the CTL group with partial AA.

### 2.3. Effect of MSCT on Skin Inflammatory Factor Gene Expression

To elucidate the observed differences in hair loss between the two groups, we investigated the expression patterns of inflammatory factors in the skin, particularly those influenced by direct application of IFN-γ and MSCs. Comparative gene expression analyses were conducted between the Healthy Control (HC) group and the saline-treated CTL AA group, as well as between the two AA-induced mouse groups (CTL group and MSC group). The HC group consisted of six female C3/HeJ mice of the same age as those in the CTL and MSC groups. These mice did not undergo induction of AA and also did not receive any treatment.

We examined the mRNA expression of immune-related cytokines, including JAK1, JAK2, STAT1, STAT3, IFN-γR, IL-1β, IL-6, IL-10, IL-15, IL-17α, IL-18, and IL-22 (Figure 4). In the AA-induced CTL mice, JAK1, JAK2, STAT1, STAT3, IFN-γR, IL-1β, IL-10, and IL-17α exhibited significant increases compared to the HC group. Conversely, in the MSC-treated AA group, JAK1, JAK2, STAT1, STAT3, IFN-γR, IL-1β, IL-10, IL-18, IL-17α, IL-15, and IL-6 showed significant decreases compared to the saline-treated CTL-AA group. Notably, the expression of IL-10 was significantly induced with IFN-γ and reversed with MSCs.

### 2.4. Effect of MSCT on Hair Cycle-Related Gene Expression

We conducted a comparative analysis of the expression levels of molecules within the Wnt/β-catenin signaling pathway, which are crucial for anagen regrowth at the chronic stage of AA-like patches (Day 70). 

In the chronic exposure to IFN-γ group (CTL group), it is observed that the mRNA of Lef1 and β-catenin exhibit higher expression compared to the HC group. FGF7 and FGF2 mRNA were also enhanced under chronic AA-like environments. Interestingly, MSCT normalized the levels of Wnt/β-catenin signaling molecules and growth factors of AA group to the levels of HC group (Figure 5).

### 2.5. Histologic Changes via MSC Treatment in AA Mice

Histological analysis of dorsal skin tissue collected on Day 70 confirmed heightened CD4+CD8+ T cell infiltration around hair follicles and subcutaneous tissue in IFN-γ-induced AA mice (both MSC group and CTL group) compared to the HC group (Figure 6). However, a reduction in T cell infiltration was notably observed in the dermis and subcutaneous fat layer of the MSC-treated group compared to the CTL group. These findings suggest that MSCT contributed to alleviating the symptoms of AA.

### 2.6. Effects of MSCT on Circulating Inflammatory Factor Regulation

In order to assess the potential effects of IFN-γ and MSCs on tissues beyond the skin, where chronic inflammatory indications were noted, we investigated the levels of inflammatory cytokine expression in serum. We utilized the BEADS array kit to analyze serum samples collected on Day 70, measuring inflammatory cytokines (Figure 7).

Overall, the inflammatory cytokines, namely IL-12p70, TNF, IFN-γ, MCP-1, IL-10, and IL-6, identified in the serum exhibited similar levels of expression across groups, with no significant differences observed.

## 3. Discussion

Our study aimed to investigate whether intralesional MSCT could demonstrate efficacy for AA in vivo. We established an IFN-induced AA mouse model, histologically corroborated with increased T-cell infiltration in the dermis and subcutaneous perifollicular area. Our hypothesis posited that local administration of MSCT to the AA lesions in a mice model could improve the microenvironment surrounding hair follicles, thus facilitating hair regrowth. Our results revealed that MSCT downregulated the AA signature proinflammatory cytokines at the molecular level and mitigated alopecia symptoms. 

Hair loss and regrowth was comparable between MSC and CTL groups for the initial 4 weeks. However, during the subsequent hair cycle initiation, the CTL AA group exhibited more extensive hair shedding compared to the MSC AA group. Importantly, MSC administration resulted in decreased AA progression or alleviation of symptoms in both diffuse and partial subtypes. 

AA is a complex, polygenic, immune-mediated condition. Efforts to discover new treatment candidates are ongoing. Numerous serum and tissue biomarkers are known for AA, including Th1 cytokines (IL-1, IL-1β, IFN-γ, TNF-α, IL-2, IL-12, IL-18, CCL4, CCL5, CXCL9, CXCL10, and CXCL11), Th2 cytokines (IL-4, IL-6, IL-10, IL-13, IL-17E, IL-31, IL-33, CCL13, CCL17, and CCL18), Th17 cytokines (IL-17α, IL-21, IL-22, IL-23, CCL20, and CCL27), Wnt/β-catenin signaling pathway-associated marker s (DKK1, β-catenin, and TCF/LEF), and other biomarkers (C-reactive protein, macrophage migration inhibitory factor, vitamin D, inducible co-stimulator molecule, matrix metalloproteinase 12, and phosphodiesterase-4) [18,19,20,21,22]. Consistently, our study showed increased expression of Th1 cytokines (IFN-γR, IL-1β), Th2 cytokines (IL-10), and Th17 cytokines (IL-17α) in the skin of AA-induced mice, while IL-18, IL-22, IL-15, and IL-6 also increased but not significantly. In addition, the mRNA expression of JAK1, JAK2, STAT1, and STAT3 (which are major pathogenic pathways in the pathogenesis of AA) were significantly increased in the skin of AA-induced mice, and this increase was reverted via MSC treatment.

Although some serum biomarkers such as IL-12, TNF-α, IFN-γ, and IL-6 are known to reflect disease severity in human AA subjects, little is known about changes in the serum in the AA mice model [22,23,24]. Byun et al. reported a decreased serum level of IFN-γ when MSCs were pretreated before AA induction in a mouse model [14]. In our study, serum IL-12p70, TNF, IFN-γ, MCP-1, IL-10, and IL-6 levels did not show any difference between groups unlike tissue expression. This made it challenging to observe the systemic effects of MSC therapy. In a mouse model of bronchiolitis obliterans, MSC infusion decreased IFN-γ level and increased IL-10 level in the serum [25]. Sun et al. reported that injection of MSC in rats with acute pancreatitis reduced serum levels of IL-6 and TNF-α [26]. Additionally, one study found that MSC therapy reduced serum IL-12 levels in both Sjögren’s syndrome patients and mice [27]. 

In this study, the MSC-treated AA group showed significantly decreased mRNA expressions of JAK1, JAK2, STAT1, STAT3, IFN-γR, IL-1β, IL-10, IL-17α, IL-18, and IL-15 in the skin compared to the saline-treated CTL AA group. Since IFN-γ and IL-15 are key players in AA pathogenesis and are therapeutic targets of JAK inhibitors, MSCs’ immunosuppressive effects on these cytokines in vivo may have therapeutic implications [28]. The role of IL-10 in MSCT is somewhat controversial. Recent studies have reported that certain populations with AA who exhibit Th2 inflammatory features, including IL-10 in the hair loss area, present more persistent disease courses [29]. In this context, the increased IL-10 observed with chronic IFN treatment in our mouse model, which reflects severe AA status, was downregulated via MSC therapy, suggesting a therapeutic implication. Conversely, other studies have shown that MSCs decrease levels of IL-1β, IL-17α, and IL-6 while increasing IL-10 levels in various autoimmune diseases [30,31]. IL-10 is not only a Th2 cytokine but also an immunosuppressive cytokine. Thus, it can be speculated that the increase in IL-10 due to MSCT could serve an immunomodulatory role or help correct Th1 and Th2 imbalance in Th1-predominant autoimmune diseases.

Despite gross and histological similarities in the hair cycle between the IFN and MSC groups, both of which were in the telogen stage at the time of skin biopsy (Day 70), significant differences were evident in the expression of Wnt/β-catenin molecules between the two groups. Contrary to our expectations, in vivo mouse skin analysis revealed that Wnt/β-catenin signaling was activated at the mRNA level via chronic IFN exposure and was reverted with MSC therapy. Specifically, the expression levels of LEF1, β-catenin, FGF7, and FGF2 mRNA were increased via IFN treatment, while the expression level of DKK1 was decreased in the skin of AA-induced mice. MSCT restored the expression levels of LEF1, β-catenin, FGF7, and FGF2 to levels similar to those of HCs. Similar results have been reported, showing that activated Wnt/β-catenin signaling, indicated by elevated β-catenin protein expression, was observed in fibroblast-like synoviocytes of rheumatoid arthritis patients [32]. Consistently, a recent study revealed that the secretome of stem cells included DKK1, which could suppress the hair cycle, and that DKK1-knockout stem cell application was beneficial for hair regeneration [33]. Thus, enhanced Wnt/β-catenin signaling might be influenced by chronic autoimmune inflammation, and this possibility needs to be confirmed through future studies. 

However, there is substantial evidence of the tissue regenerative effects of MSCT in various tissues, including hair follicles, related to the activation of Wnt/β-catenin signaling. We previously reported that MSCs promote hair growth in a hair follicle culture model under IFN conditions by activating Wnt/β-catenin signaling and stimulating the secretion of various growth factors [17]. 

Yan et al. found that miR-22-5p in dermal papilla cell-derived exosomes can regulate hair follicle stem cell proliferation by targeting LEF1, a key transcription factor in the Wnt signaling pathway [34]. Hu et al. reported that miR-218-5p in dermal papilla spheroid-derived exosomes inhibits the expression of the Wnt signaling inhibitor SFRP2, thereby upregulating β-catenin and promoting hair regeneration [35]. In this study, we observed less hair shedding in MSC-treated AA mice compared to saline-treated AA mice. However, the expression of Wnt/β-catenin signaling molecules was more suppressed via MSCT at the mRNA level. This disparity may stem from differences between the in vivo hair cycle and in vitro conditions or the duration of IFN treatment (long-term vs. short-term). Building upon our findings, it can be inferred that the effects of MSCT on Wnt/β-catenin molecules in vivo are complex and warrant further investigation for clinical application. 

A previous study demonstrated the inhibitory roles of MSCs in the development of AA in a mouse model [14]. They induced AA using skin grafts from AA-developed mice to other individuals and administered MSCs intravenously. Differences in the timing and route of MSC administration, the type of MSCs used, or the method of AA induction may result in varying therapeutic outcomes. However, we hypothesize that preventing AA development with MSCs might be a more effective strategy than treating established AA with MSCs. When applied to humans, intravenous administration of MSCs requires a larger number of cells and presents efficiency and safety issues, such as MSCs getting trapped in the lungs [36]. Therefore, intralesional injection of MSCs is considered a practical alternative for future human AA treatments. Local administration of MSCs could theoretically be more efficient than systemic administration, particularly in targeting localized inflammation. The beneficial effects of local MSC administration in the skin of patients with atopic dermatitis have already been reported [37].

The long-term safety of JAK inhibitors in AA treatment is not established, and recurrence after discontinuation is frequently reported [38]. This study provides valuable information on the in vivo effects and mechanisms of MSC therapy, offering insights for clinical translation. The results suggest that combining MSC therapy with other treatments or using MSCs as sequential maintenance therapy after achieving remission with JAK inhibitors could enhance therapeutic outcomes and help prevent AA recurrence.

This study acknowledges several limitations. The use of an AA mouse model necessitates confirmation of these findings in human AA-affected scalp areas. MSCT responses may vary due to donor factors, even within the same individual. The outcome of MSCT faces challenges such as reproducibility, high costs, and long preparation periods. However, MSCT offers advantages, including no cytotoxicity and lasting immunomodulatory effects, making it potentially ideal for personalized medicine [39]. Additionally, further research is needed to identify specific molecules with anti-inflammatory and immunomodulatory actions secreted by MSCs.

Our study provides clinically relevant insights, showing that local administration of MSCs to AA sites significantly reduces inflammation and prevents further hair loss. However, much work remains to apply MSCs to human subjects with AA. Future studies should determine the effective dosage, interval, and administration route of MSCs to optimize therapeutic outcomes. 

## 4. Materials and Methods

### 4.1. Animals and Human Bone Marrow-Derived Mesenchymal Stem Cells (hMSCs)

Five-week-old female C3/HeJ mice were acquired from Japan SLC, Inc. (Shizuoka, Japan). The experimental procedures adhered to institutional guidelines and were approved by the Institutional Animal Care and Use Committee (Approval No. EPSMH20190000FA) of Eunpyeong St. Mary’s Hospital. The mice were housed in a facility with a 12 h light–dark cycle. hMSCs from a healthy donor were sourced from the Catholic Institute of Cell Therapy (CIC, Seoul, Republic of Korea; IRB: PC16TISE0029). hMSCs were sub-cultured every 3 days in low-glucose Dulbecco’s Modified Eagle Medium (Gibco BRL, Life Technologies, Karlsruhe, Germany) supplemented with 10% fetal bovine serum (FBS; Gibco BRL, Life Technologies, Karlsruhe, Germany) and 1% penicillin/streptomycin (Gibco BRL, Life Technologies, Karlsruhe, Germany). The hMSCs passage number utilized in the experiment was 6.

### 4.2. Induction of AA

After shaving the hair on the back of 18-week-old female C3/HeJ mice (purchased from Central Lab. Animal, Japan SLC) using clippers, any remaining hair was removed with a depilatory cream. In the initial week, the mice received intraperitoneal (IP) injections of recombinant murine IFN-γ (Peprotech, Cranbury, NJ, USA) at a concentration of 2 × 10^4^ U/mL, totaling 100 μL, administered five times weekly (Figure 8). During the second and third weeks, intradermal (ID) injections of IFN-γ at the same concentration, totaling 400 μL per animal, were evenly given at eight sites (50 μL each site) of dorsal skin, three times weekly. Three weeks after AA induction, confirmation of AA development was achieved through skin biopsy of 3 sacrificed mice with hair loss when increased infiltration of CD4+CD8+ T cells in the dermis and subcutaneous perifollicular area was seen. Following AA induction completion, IFN-γ was administered intraperitoneally three times weekly to sustain AA throughout the experiment. If the AA induction was deemed insufficient, the induction cycle was repeated once more.

### 4.3. Administration of hMSCs 

Diluted cultured hMSCs (obtained from the Catholic Institute of Cell Therapy, Seoul, Republic of Korea) at a concentration of 1 × 10^6^ cells were intradermally injected at 50 μL each into 8 dorsal sites. One week later, the intradermal injection of hMSCs was repeated. The control group, not treated with hMSCs, received an equivalent volume of saline injected using the same method. 

### 4.4. Image Analysis 

The area of hair loss and the area with hair were compared using photographs taken weekly from day 0 to 10 weeks. After isolating the two regions of differing color, Adobe Photoshop (Version CC 2019, Adobe, San Jose, CA, USA) and Microsoft Excel (Version 2407, Microsoft, Redmond, WA, USA) were utilized to convert the areas into numerical values for comparison.

### 4.5. Mouse Tissue Sample Analysis

Mice were sacrificed on Day 70, seven weeks after the initial MSC administration, to collect skin. Skin samples were obtained from the CTL and MSC group. Skin samples were also collected from six age-matched HC mice that did not receive any treatment. These samples were used as a baseline for comparing inflammatory cell infiltration and inflammatory markers with those from the CTL and MSC groups. For histological examination, the dorsal skin was biopsied and fixed with 4% paraformaldehyde, followed by embedding the tissue into paraffin blocks and cutting it into 5 µm sections. After attaching the sections to the silane-coated slides, the slides were deparaffinized with xylene and ethanol, and then were boiled in antigen retrieval buffer (Novus Biologicals, Centennial, CO, USA) using a microwave. Next, the slides were blocked using SuperBlock blocking buffer (Thermo scientific, Rockford, IL, USA) for 30 min to prevent non-specific antibody binding. After blocking, slides were immune stained with an anti-CD4 antibody (1:100, BD Bioscience, San Jose, CA, USA) and anti-CD8 antibody (1:100, Santa Cruz Biotechnology, Dallas, TX, USA) overnight at 4 °C. Then, the slides were washed with phosphate buffer saline and incubated with an anti-rat AlexaFluor-488 and anti-rat AlexaFluor-594 secondary antibody (1:200, Abcam, Fremont, CA, USA) at room temperature for 2 h. The slides were stained with DAPI mounting solution (Vector Laboratories, Newark, CA, USA) and were observed under a microscope (200×) for immunofluorescence staining.

For total RNA isolation, total RNA isolation followed the manual protocol using Trizol reagent (Invitrogen, Carlsbad, CA, USA), and the extracted RNA was then synthesized into cDNA using the MG cDNA Synthesis Kit (CancerROP, Seoul, Republic of Korea) for RT-PCR analysis. Primer sequences for gene expression are provided in Table 1, and the primers were synthesized by Bioneer (Bioneer, Daejeon, Republic of Korea). 

On the same day, serum was separated from blood collected via orbital extraction, and a cytokine bead array was conducted using the Mouse Inflammation CBA Kit (BD Biosciences, San Jose, CA, USA). Data were obtained using a FACSCanto II flow cytometer (BD Biosciences, San Jose, CA, USA), and the analysis was conducted following the manufacturer’s instructions.

### 4.6. Statistical Analysis

All statistical analyses were conducted using GraphPad Prism (Version 8, GraphPad Software, San Diego, CA, USA). The Mann–Whitney test was used to compare improvements in proportion of hair-bearing area between the CTL group and the MSC group. In the rest of the analyses except for this analysis, significant differences were assessed at *p* < 0.05 using Student’s *t*-test and one-way analysis of variance (ANOVA). All tests were one-sided.

## Figures and Tables

**Figure 1 ijms-25-09236-f001:**
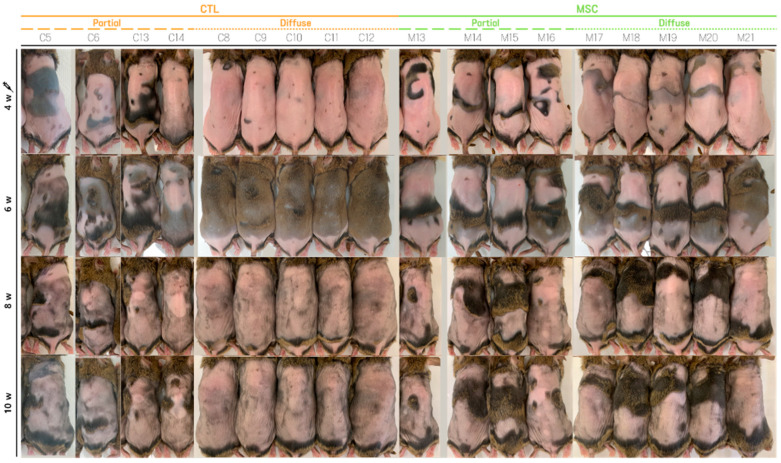
Gross images of the dorsal skin of 18 representative Alopecia Areata (AA)-induced mice at week 4, 6, 8, and 10. Mice C5, C6, C13, and C14 belong to the control (CTL) group with partial AA (areas with more than 25% hair coverage), and mice C8-C12 belong to the CTL group with diffuse AA (areas with less than 25% hair coverage). Mice M13–M16 belong to the Mesenchymal Stem Cell (MSC) group with partial AA, and mice M17-M21 belong to the MSC group with diffuse AA. At week 4, extensive AA patches were observed in both groups. In the MSC group, diluted cultured Human Bone Marrow-Derived Mesenchymal Stem Cells (hMSCs) were intradermally injected into eight dorsal sites at week 4 and 5, while saline was injected in the CTL group during the same period. By week 6, hair regrowth was evident in most mice from both groups. However, hair loss recurred during the subsequent hair cycle, with prominent hair loss observed in most mice from both groups at week 8 and continuing through week 10. In fact, overall, the MSC group showed less hair loss compared to the CTL group.

**Figure 2 ijms-25-09236-f002:**
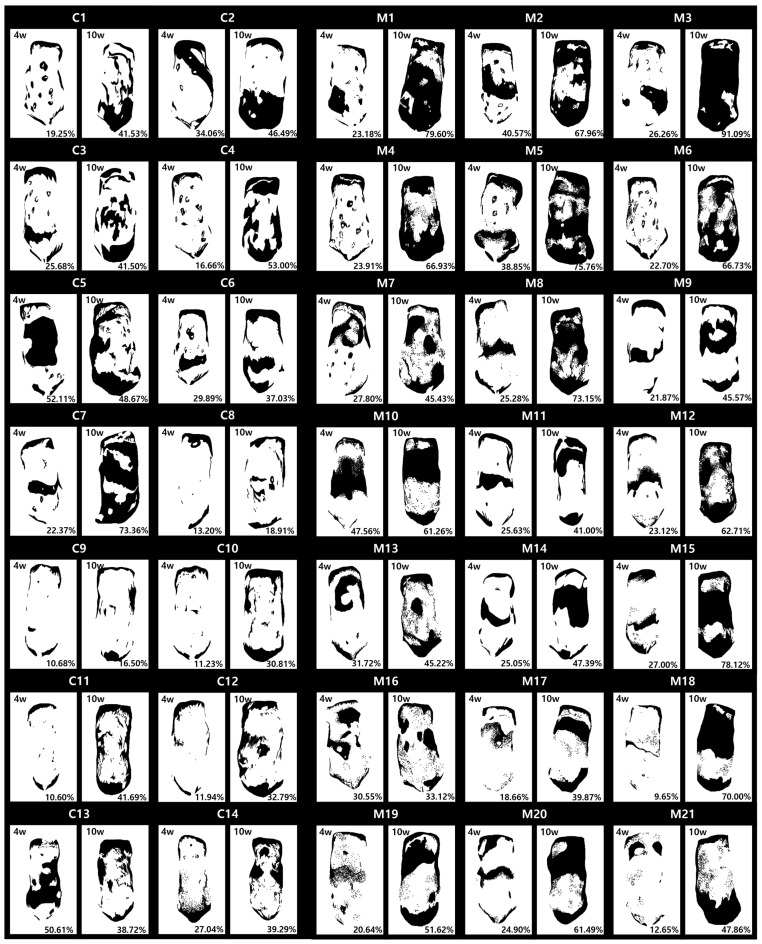
Image showing the part with hair (black) and the part without hair (white). For each mouse at week 4 and 10, the proportion of areas with hair was calculated using pixels and the resulting values were noted. C: CTL group, M: MSC group.

**Figure 3 ijms-25-09236-f003:**
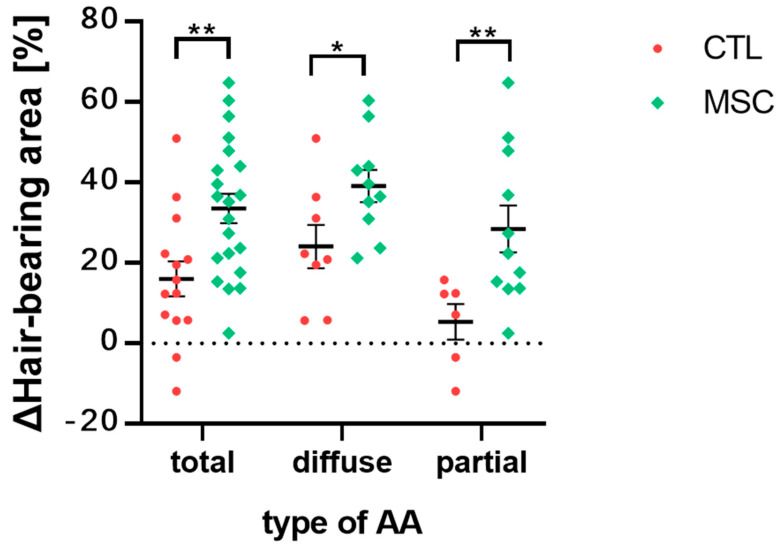
Comparative analysis of changes in the hair-bearing area ratio between the CTL and MSC groups. All mice from the CTL and MSC group were included, with a total of 14 mice in the CTL group and 21 mice in the MSC group. For the diffuse subtype, 8 mice were in the CTL group, and 10 were in the MSC group, while for the partial subtype, 6 mice were in the CTL group, and 11 were in the MSC group. The degree of improvement in the proportion of hair-covered areas between week 4 and week 10 was calculated for each mouse. The MSC group exhibited significantly greater hair regrowth compared to the CTL group. Additionally, when comparing the diffuse AA subtypes between the two groups, the MSC group showed superior results, as did the MSC group with partial AA subtypes. The data represent the means ± SEM; n = 3; results were statistically significant at * *p* < 0.05 and ** *p* < 0.01.

**Figure 4 ijms-25-09236-f004:**
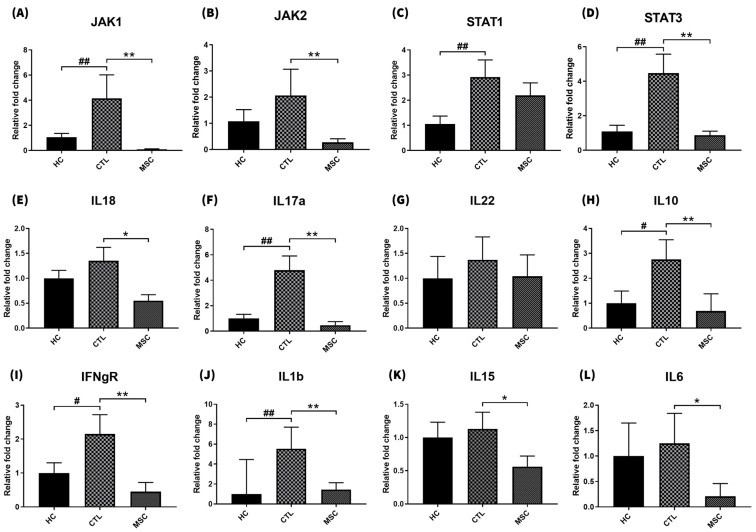
Compared relative fold changes in mRNA expression of pro-inflammatory/anti-inflammatory cytokines in mice dorsal skin. The mRNA expression levels of (**A**) JAK1, (**B**) JAK2, (**C**) STAT1, (**D**) STAT3, (**E**) IL-18, (**F**) IL-17α, (**G**) IL-22, (**H**) IL-10, (**I**) IFN-γR, (**J**) IL-1β, (**K**) IL-15, and (**L**) IL-6 in the Healthy Control (HC), CTL, and MSC groups were measured and analyzed. JAK1, JAK2, STAT1, STAT3, IFN-γR, IL-1β, IL-10, and IL-17α were significantly increased in the CTL AA group compared to the HC group. JAK1, JAK2, STAT1, STAT3, IFN-γR, IL-1β, IL-10, IL-18, IL-17α, IL-15, and IL-6 were significantly decreased in the MSC AA group compared to the CTL AA group. The data represent the means ± SEM; n = 3. Results are statistically significant at # *p* < 0.05 and ## *p* < 0.01 compared to the HC group and * *p* < 0.05 and ** *p* < 0.01 compared to the CTL group.

**Figure 5 ijms-25-09236-f005:**
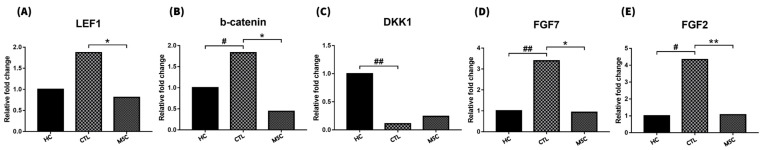
Compared relative fold changes in mRNA expression of genes within the Wnt/β-catenin pathway (a major signaling pathway in the hair cycle) in mice dorsal skin. The mRNA expression levels of (**A**) LEF1, (**B**) β-catenin, (**C**) DKK1, (**D**) FGF7, and (**E**) FGF2 in HC, CTL, and MSC groups were measured and analyzed. β-catenin, FGF7, and FGF2 were significantly increased in the saline-treated CTL AA group compared to the HC group. LEF1, β-catenin, FGF7, and FGF2 were significantly decreased in the MSC-treated AA group compared to the saline-treated CTL AA group. (**A**–**E**). The data represent the means ± SEM; n = 3; results are statistically significant at # *p* < 0.05 and ## *p* < 0.01 compared to the HC group, and * *p* < 0.05 and ** *p* < 0.01 compared to the CTL group.

**Figure 6 ijms-25-09236-f006:**
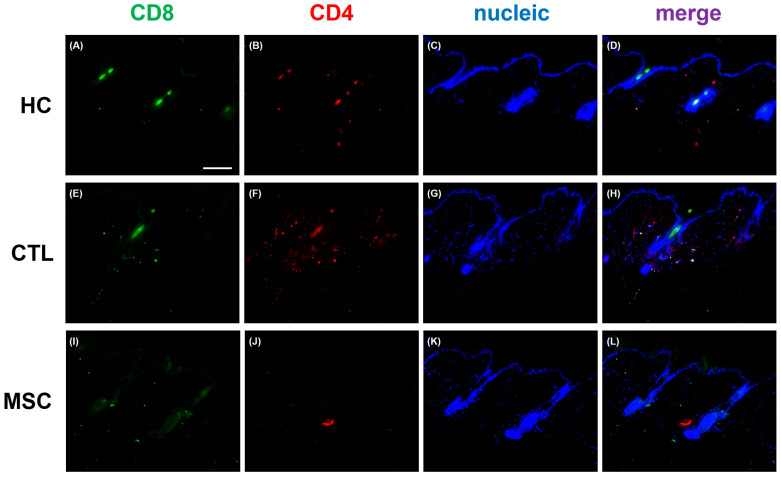
Histological analysis of mice dorsal skin tissue. Immunofluorescence staining was performed on tissues on Day 70 to compare immune cell infiltration between groups. The images presented include: (**A**) CD8, (**B**) CD4, (**C**) nucleic, and (**D**) merged of the HC group; (**E**) CD8, (**F**) CD4, (**G**) nucleic, and (**H**) merged of the CTL group; and (**I**) CD8, (**J)** CD4, (**K**) nucleic, and (**L**) merged of the MSC group. CD8 expression is shown in green, CD4 in red, and nucleic in blue. Compared to the HC group, the infiltration of CD4+CD8+ T cells in the hair follicles and surrounding subcutaneous tissue is increased in the CTL group. This increased infiltration is reverted in the MSC group. However, MSC treatment (MSCT) confirmed a decrease in CD4+CD8+ T cells in the dermis and subcutaneous fat layer compared to CTL group. Scale bar = 100 μm.

**Figure 7 ijms-25-09236-f007:**
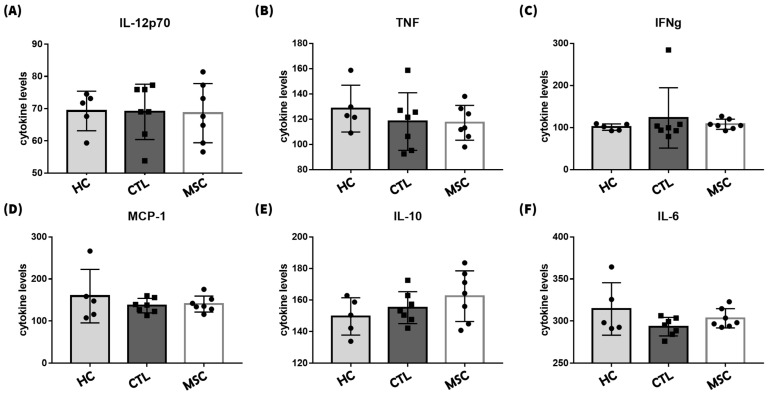
Comparison of expression of inflammatory cytokines in serum. We analyzed serum using the BEADS array kit to measure inflammatory cytokines in blood drawn on Day 70. The expression levels of (**A**) IL-12p70, (**B**) TNF, (**C**) IFN-γ, (**D**) MCP-1, (**E**) IL-10, and (**F**) IL-6 in serum. There were no significant differences between groups. The data represent the means ± SEM; n = 3.

**Figure 8 ijms-25-09236-f008:**
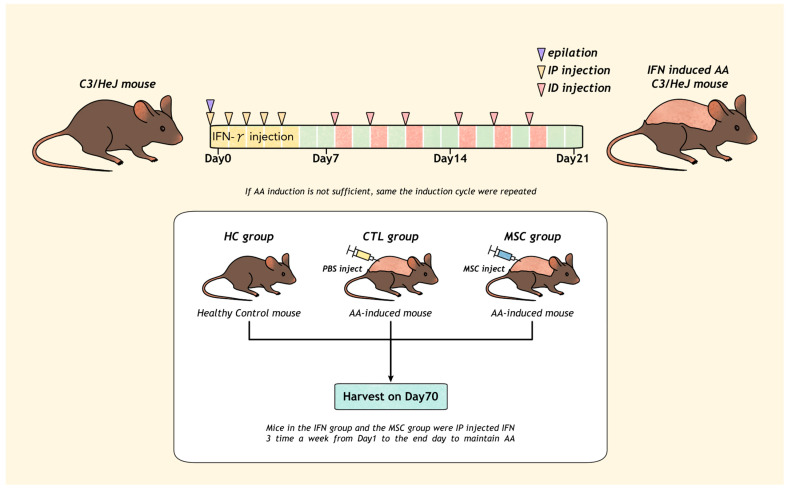
Induction of AA in C3/HeJ mouse via Interferon-gamma (IFN-γ) injections. Daily intraperitoneal administration of IFN-γ on Day 1–5 and intradermal administration of IFN-γ on Day 8, 10, 12, 15, 17, and 19 (3 times a week for 2 weeks) were conducted to develop AA. AA-induced mice were divided into two groups: CTL group, which received saline administration, and MSC group, which received MSC administration.

**Table 1 ijms-25-09236-t001:** Primer sequences for gene expression.

Gene	Primer	Sequence (5′ to 3′)	Length
*JAK1*	Forward	TAT TGG GTG GCC AGA AGC AG	20
	Reverse	GTG GTT CAT GAG GTC TCG CA	20
*JAK2*	Forward	GAG GTG GTC GCT GTG AAG AA	20
	Reverse	GCA CTG TAG CAC ACT CCC TT	20
*STAT1*	Forward	TTC AGC AGC TGG ACT CCA AG	20
	Reverse	ACG AGA CAT CAT AGG CAG CG	20
*STAT3*	Forward	AAC GAC CTG CAG CAA TAC CA	20
	Reverse	TCC ATG TCA AAC GTG AGC GA	20
*IFNγR*	Forward	GCA ATG ACC CAA GAC CAG TG	20
	Reverse	ACT GTG AAT GGG TGT GGG AA	20
*IL-10*	Forward	CGG GAA GAC AAT AAC TGC ACC C	22
	Reverse	CGG TTA GCA GTA TGT TGT CCA GC	23
*IL-15*	Forward	CAT TTT GGG CTG TGT CAG TG	20
	Reverse	GCA ATT CCA GGA GAA AGC AG	20
*IL-17α*	Forward	CAG ACT ACC TCA ACC GTT CCA C	22
	Reverse	TCC AGC TTT CCC TCC GCA TTG A	22
*IL-18*	Forward	CAG GCC TGA CAT CTT CTC CAA	21
	Reverse	CTG ACA TGG CAG CCA TTG T	19
*IL-1β*	Forward	TGG ACC TTC CAG GAT GAG GAC A	22
	Reverse	GTT CAT CTC GGA GCC TGT AGT G	22
*IL-22*	Forward	GCT TGA GGT GTC CAA CTT CCA G	22
	Reverse	ACT CCT CGG AAC AGT TTC TCC C	22
*IL-6*	Forward	TAC CAC TTC ACA AGT CGG AGG C	22
	Reverse	CTG CAA GTG CAT CAT CGT TGT TC	23
*DKK1*	Forward	ATA TGC ATG CCC TCT GAC CA	20
	Reverse	CGG AGC CTT CTT GTC CTT TG	20
*FGF2*	Forward	CGA CCC ACA CGT CAA ACT ACA	21
	Reverse	GTA ACA CAC TTA GAA GCC AGC A	22
*FKF7*	Forward	ATA GAA ACA GGT CGT GAC AAG G	22
	Reverse	CAG ACA GCA GAC ACG GAA C	19
*LEF1*	Forward	GTC GAC TTC AGG TGG TAA GAG A	22
	Reverse	TGC TGT CAG TGT TCC TTG GG	20
*β-catenin*	Forward	GGC AGC GGC AGG ATA CAC GG	20
	Reverse	CAG GAC ACG AGC TGA CGC GG	20

## Data Availability

The data that support the findings of this study are available from the corresponding author upon reasonable request.

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
