# Peer review of "Mesenchymal Stem Cell Therapy in Alopecia Areata: Visual and Molecular Evidence from a Mouse Model"

_ijms, 2024, doi:10.3390/ijms25179236_

Round 1
Reviewer 1 Report
Comments and Suggestions for Authors
The authors performed a study about Mesenchymal Stem Cell Therapy in Alopecia Areata. The article is of interest however some changes are needed:
- Can you specify better the type/phenotype (universal or localized) of induced of AA in mice?
- Did you find any different response to the treatment in mice with universal or localized AA?
- Did you test also cytokines that follow the line JAK1/JAK2 since JAK inhibitors are now the gold standard for the treatment of AA?
- Can you explain why a clinician should prefer this treatment to anti-JAK therapies?
Author Response
Comments 1: Can you specify better the type/phenotype (universal or localized) of induced of AA in mice?
Response 1: Thank you for your valuable comments. As highlighted in Figures 1 and 2, alopecia areata (AA) became apparent in most subjects by week 4. We defined diffuse AA as cases where the hair-bearing area of the dorsal skin was less than 25%, and partial AA as cases where it was more than 25% at week 4.
In the updated Figure 2, among the 14 mice in the CTL group, 8 mice (C1, C4, C7, C8–C12) exhibited diffuse hair loss, while the remaining 6 mice (C2–C3, C5–C6, C13–C14) showed partial hair loss at week 4. Similarly, among the 21 mice in the MSCT group, 10 mice (M1, M4, M6, M9, M12, M17–M21) had diffuse hair loss, and the remaining 11 mice (M2–M3, M5, M7–M8, M10–M11, M13–M16) exhibited partial hair loss at week 4.
(Page 3, 2nd paragraph, Line 104-109)
Comments 2: Did you find any different response to the treatment in mice with universal or localized AA?
Response 2: Thank you for pointing this out. We calculated the hair-bearing surface area ratio (%) relative to the total dorsal skin area using ImageJ for all mice, allowing us to quantitatively compare the therapeutic effects between groups. We assessed the degree of improvement in the hair-bearing area (%) between week 4 (baseline) and week 10 for each mouse (as shown in the updated Figure 2).
Regardless of the type of hair loss, all mice in the MSC group showed an improvement in the hair-bearing area (%) at week 10 compared to week 4. Notably, the increase in the hair-bearing area (%) at week 10 was significantly higher in the MSC-treated group compared to the saline-treated control group (Figure 3). Contrary to our expectation that mice with diffuse AA might show a poorer treatment response, those with diffuse AA in the MSC group demonstrated an average improvement of 39.11%, slightly higher than the 28.48% improvement observed in mice with partial AA in the same group. However, this difference was not statistically significant. In contrast, the degree of improvement in hair loss was significantly greater in mice with diffuse AA compared to those with partial AA in the control group (p<0.05).
(Page 3, 2nd paragraph, Line 102-104, 109-113)
Comments 3: Did you test also cytokines that follow the line JAK1/JAK2 since JAK inhibitors are now the gold standard for the treatment of AA?
Response 3: We additionally examined the mRNA expression of JAK1, JAK2, STAT1, STAT3 in the skin. JAK1, JAK2, STAT1, STAT3 were significantly increased in the CTL group compared to the HC group, and were significantly decreased in MSC group compared to CTL group.
(Page 6, 2nd paragraph, Line 156-161 & Page 7, Figure 4)
Comments 4: Can you explain why a clinician should prefer this treatment to anti-JAK therapies?
Response 4: Thank you for pointing this out. JAK inhibitors are the first FDA-approved treatment for AA, and it is clear that clinicians should prioritize therapies that have proven efficacy and safety. However, approximately 30% of patients do not respond to anti-JAK therapies. Moreover, the long-term safety of JAK inhibitors in AA treatment has yet to be established, and recurrence after discontinuation or dose reduction is frequently reported. JAK inhibitors function primarily by targeting the intracellular kinase JAK and blocking the JAK-STAT signaling pathway.
In contrast, MSCs not only exert immunosuppressive effects on the cytokines involved in AA pathogenesis, but also activate the Wnt/β-catenin signaling pathway and stimulate growth factors that are crucial for hair cycle regulation. From this, we may anticipate potential hair growth effects in some refractory AA patients. We believe that combining MSC therapy with other treatments or using MSCs as a sequential maintenance therapy after achieving remission with JAK inhibitors could enhance therapeutic outcomes and help prevent AA recurrence.
(Page 2, 1st paragraph, Line 48-53 & Page 11, 2nd paragraph, Line 318-323)
Reviewer 2 Report
Comments and Suggestions for Authors
Dear author,
This study aimed to using an interferon-γ (IFN-γ)-induced AA mouse model to verify the efficacy of topically administered Mesenchymal Stem Cells (MSCs) therapy in the treatment of Alopecia Areata (AA). The authors concluded that MSC-treated group showing (1) slightly milder hair loss (2) longer, thicker hair growth and (3) reduced local inflammatory cytokines compared to the control group. Figures 1A and 1B show the changes in the area of Alopecia Areata after MSCs treatment. However, I do not observe a significant difference in the AA area between the two groups as depicted in Figure 1. Additionally, the “outer boundary of the epilated spots shifted inward more noticeably” proposed by the authors is not clearly reflected in the figure.
Regarding the statistical results, I would like to ask the authors to clarify whether C1-C6 and M1-M6 in Figures 1E and 1F represent 6 mice in each group. I noticed that lines 81-86 in the text state that “21 mice were allocated to the MSCT group”, but the statistical data here only include 6 mice each for the control and MSCT groups, which does not seem sufficient to represent the overall situation.
The same, line 123-125, the authors proposed that “Particularly, the MSC-treated group displayed longer, thicker hair growth with fewer broken hairs compared to the saline-treated CTL group”. However, I really cannot observe any difference through Figure2. Since mouse hair is short and dense, Folliscope technology is not suitable for observing mouse hair. To observe the length and density of mouse hair more accurately, you can consider the following methods: (1) Use a high-magnification dissecting microscope to take pictures. A regular dissecting microscope can magnify 40X, using a 2X objective can further magnify the hair growth on the back of the mouse to 80-100X; (2) Take a biopsy and prepare sections. H&E staining can be used to detect the density of hair follicles; (3) Pluck a small amount of hair and observe the thickness under the microscope, then compile statistical data. Based on the figures provided by the authors, I really cannot conclude that the hair growth in the MSC-treated group is longer, denser, or less prone to breakage.
The above two points are my biggest concerns. In addition, there are several small points that need to be improved.
(1) Line84-85 “Confirmation of AA was based on histological analysis revealing infiltration of CD4+CD8+ T cells”. Could author provide more details and necessary figures for this?
(2) Line136 introduced a “healthy control group”, but there was not much explanation in the Materials and Methods. Did the healthy control group receive the same CTL treatment?
(3) The immunostaining results for CD4+CD8+ T cell infiltration in Figure 5 do not show significant differences. The authors could consider the following improvements: (a) use arrows to indicate the location of positive signals, (b) ensure that the immunofluorescence staining is clearer, and (c) include statistical results to better present the findings.
(4) Line 203-204 “Our results revealed that MSCT downregulated the AA signature proinflammatory cytokines at the molecular level and visibly mitigated alopecia symptoms”. I don't think it's visibly. The following paragraph, lines 205-213, needs careful revision. The results presented in the figure are not as positive as described in this paragraph.
(5) In the Materials and Methods section, 4.2 and 4.3 mention that during the induction process of AA, IFN-γ was administered at eight sites, and during treatment, hMSCs were administered at eight sites. Could the authors provide detailed information about these eight sites? This information is crucial for accurate phenotype observation.
Sincerely,
Author Response
Comments 1: This study aimed to using an interferon-γ (IFN-γ)-induced AA mouse model to verify the efficacy of topically administered Mesenchymal Stem Cells (MSCs) therapy in the treatment of Alopecia Areata (AA). The authors concluded that MSC-treated group showing (1) slightly milder hair loss (2) longer, thicker hair growth and (3) reduced local inflammatory cytokines compared to the control group. Figures 1A and 1B show the changes in the area of Alopecia Areata after MSCs treatment. However, I do not observe a significant difference in the AA area between the two groups as depicted in Figure 1. Additionally, the “outer boundary of the epilated spots shifted inward more noticeably” proposed by the authors is not clearly reflected in the figure.
Response 1 : Thank you for your insightful comments. We completely agree and have made the necessary revisions. We removed the existing Figure 1 and added three new figures (Figures 1–3).
Figure 1 shows the gross images of the dorsal skin of all 35 AA-induced mice from week 0 (start of induction) to week 10 (sacrifice). To facilitate a clearer comparison of visible changes, the affected areas were highlighted: AA-affected regions were marked in white, and areas with hair were highlighted in black at weeks 4 and 10 for each mouse (Figure 2). We then calculated the proportion of areas with hair by pixel count. Cases where the hair-bearing area was less than 25% were defined as diffuse AA, while cases with more than 25% were defined as partial AA. Using the hair-bearing area proportions from Figure 2, we calculated the differences of hair-bearing area extent (%) between week 4 and week 10 for each mouse. These differences were then compared between the CTL group and the MSC group (Figure 3).
The MSC group showed significant improvement in hair regrowth compared to the CTL group at week 10. For a more detailed analysis, further comparisons were made between the AA subtypes within each group. Notably, the MSC group with diffuse AA showed significant improvement in hair regrowth compared to the CTL group with diffuse AA at week 10, as did the MSC group with partial AA compared to the CTL group with partial AA.
(Page 3, 2nd paragraph, Line 102-106 & Page 3, 3rd paragraph, Line 114-121 & Page 4, Figure 1 & Page 5, Figure 2 & Page 6, Figure 3)
Comments 2: Regarding the statistical results, I would like to ask the authors to clarify whether C1-C6 and M1-M6 in Figures 1E and 1F represent 6 mice in each group. I noticed that lines 81-86 in the text state that “21 mice were allocated to the MSCT group”, but the statistical data here only include 6 mice each for the control and MSCT groups, which does not seem sufficient to represent the overall situation.
Response 2: Thank you for the comments. We have replaced Figures 1 and 2 to include the experimental results for all mice. The new Figures 1 and 2 present the sequential data for all 35 AA-induced mice, including the 21 mice in the MSCT group (M1–M21) and the 14 mice in the CTL group (C1–C14).
(Page 2, 7th paragraph, Line 88-90 & Page 4, Figure 1 & Page 5, Figure 2)
Comments 3: The same, line 123-125, the authors proposed that “Particularly, the MSC-treated group displayed longer, thicker hair growth with fewer broken hairs compared to the saline-treated CTL group”. However, I really cannot observe any difference through Figure2. Since mouse hair is short and dense, Folliscope technology is not suitable for observing mouse hair. To observe the length and density of mouse hair more accurately, you can consider the following methods: (1) Use a high-magnification dissecting microscope to take pictures. A regular dissecting microscope can magnify 40X, using a 2X objective can further magnify the hair growth on the back of the mouse to 80-100X; (2) Take a biopsy and prepare sections. H&E staining can be used to detect the density of hair follicles; (3) Pluck a small amount of hair and observe the thickness under the microscope, then compile statistical data. Based on the figures provided by the authors, I really cannot conclude that the hair growth in the MSC-treated group is longer, denser, or less prone to breakage.
Response 3: We completely agree that Folliscope technology is not suitable for assessing mouse hair. Since H&E staining was only available for a few mice (primarily to confirm the development of AA) at week 4, we were unable to perform a histologic comparison between week 4 and week 10. Therefore, we adopted alternative methods to demonstrate the efficacy of MSCT.
As a result, we have removed the existing Figure 1 and 2, and revised a new Figure 1 and 2, which presents gross images of the mice’s dorsal skin to illustrate the therapeutic effects of MSCT. Additionally, the new Figure 2 illustrates the areas with hair (highlighted in black) and the areas without hair (highlighted in white), along with the percentage of the hair-covered area at week 4 (immediately after AA induction) and week 10 (after two hair cycles). This enables a quantitative comparison of the therapeutic effects on hair loss between the groups.
(Page 4, Figure 1 & Page 5, Figure 2)
Comments 4: Line84-85 “Confirmation of AA was based on histological analysis revealing infiltration of CD4+CD8+ T cells”. Could author provide more details and necessary figures for this?
Response 4: Thank you for highlighting this. At week 4, among all mice with hair loss, 3 were selected for histologic confirmation. These mice were sacrificed for histological analysis of the hair loss sites. As shown in the figure on the below, we observed increased CD4+CD8+ T cell infiltration in the dermis and subcutaneous perifollicular areas, as well as dystrophic hair follicles. These findings confirmed the induction of AA. You can observe denser CD4+CD8+ T cell infiltration at week 4 (the acute stage of AA) compared to week 10 (the chronic stage of AA), as shown in the original Figure 6.
(Page 2, 7th paragraph, Line 84-88)

Comments 5: Line136 introduced a “healthy control group”, but there was not much explanation in the Materials and Methods. Did the healthy control group receive the same CTL treatment?
Response 5: Thank you for pointing this out. The healthy control group consisted of six female C3/HeJ mice of the same age as those in the CTL and MSC groups. These mice did not undergo AA induction and did not receive any treatment. We added more explanation both in the results and method section.
(Page 6, 1st paragraph, Line 152-155)
Comments 6: The immunostaining results for CD4+CD8+ T cell infiltration in Figure 5 do not show significant differences. The authors could consider the following improvements: (a) use arrows to indicate the location of positive signals, (b) ensure that the immunofluorescence staining is clearer, and (c) include statistical results to better present the findings.
Response 6: Since week 10 represents the chronic stage of AA, CD4+CD8+ T cell infiltration is less pronounced, making it difficult to discern significant differences between groups. To clarify this, we performed immunofluorescence staining, and the resulting IF images have replaced the original H&E figure. A reduction in CD4+CD8+ T cell infiltration was observed in the dermis and subcutaneous tissue of the MSC group compared to the CTL group. Thank you for your advice.
(Page 8, Figure 6 & Page 13, 1st paragraph, Line 399-401)
Comments 7: Line 203-204 “Our results revealed that MSCT downregulated the AA signature proinflammatory cytokines at the molecular level and visibly mitigated alopecia symptoms”. I don't think it's visibly. The following paragraph, lines 205-213, needs careful revision. The results presented in the figure are not as positive as described in this paragraph.
Response 7: We appreciate your feedback. We have removed the previous Figures 1 and 2 and the Folliscope imaging results, and replaced them with new Figures 1–3.
Figure 2 highlights areas affected by AA in white and hair-bearing areas in black for each mouse at weeks 4 and 10. We calculated the proportion of hair-bearing areas and classified them as diffuse AA (less than 25%) or partial AA (more than 25%).
Figure 3 shows that the MSC group exhibited significant improvement in hair regrowth compared to the CTL group at week 10. Additionally, within each group, the MSC treatment led to notable improvement in both diffuse and partial AA subtypes. In conclusion, MSC administration reduced AA progression and alleviated symptoms in both subtypes.
(Page 3, 2nd paragraph, Line 102-106 & Page 3, 3rd paragraph, Line 114-121, Page 4, Figure 1 & Page 5, Figure 2 & Page 6, Figure 3)
Comments 8: In the Materials and Methods section, 4.2 and 4.3 mention that during the induction process of AA, IFN-γ was administered at eight sites, and during treatment, hMSCs were administered at eight sites. Could the authors provide detailed information about these eight sites? This information is crucial for accurate phenotype observation.
Response 8: Thank you for your excellent points. We aimed to divide the dorsal skin into eight equal sections and administer the injections evenly to avoid concentration in one area. If a wound remained unhealed at the site of a previous injection, a new injection was given nearby. The specific injection sites are evident in some mice in the Figure 1, as they show rapid hair growth or localized hair loss, which we attribute to potential wound healing effects.
(Page 4, Figure 1 & Page 12, 1st paragraph, Line 356-359)
Round 2
Reviewer 1 Report
Comments and Suggestions for Authors
The authors improved the manuscript and the article can be accepted.
Author Response
Thank you very much for taking the time to review this manuscript.
Reviewer 2 Report
Comments and Suggestions for Authors
Dear Editor,
I have carefully reviewed the revised version of the manuscript and am pleased to see that the authors have made significant improvements based on my previous suggestions. The quality of the figures has been enhanced, the Materials and Methods section is more detailed, and the descriptions of the results are now more comprehensive and clear, making the manuscript much easier to read. Most importantly, all figures in the revised manuscript now effectively support the conclusions, leaving no room for doubt.
I do have a minor suggestion regarding Figure 1, which appears a bit overcrowded. It might be beneficial for the authors to select a few representative images for inclusion in Figure 1 and move the remaining images to the supplementary material.
Overall, I believe the manuscript has improved significantly and is suitable for publication.
Best regards,
Ruiqi
Author Response
Comment 1: I do have a minor suggestion regarding Figure 1, which appears a bit overcrowded. It might be beneficial for the authors to select a few representative images for inclusion in Figure 1 and move the remaining images to the supplementary material.
Response 1: Thank you for pointing this out. We have fully agreed and revised it. We moved Figure 1 to Supplementary Figure S1, and added a new Figure 1 showing the 9 representative mice in each group. (Page 2, 7th paragraph, Line 90-91 & Page 3, 1st paragraph, Line 102 & Page 3, Figure 1 with legend & Page 14, Supplementary Materials)